# The formation of the light-sensing compartment of cone photoreceptors coincides with a transcriptional switch

Janine M Daum[1,2], Özkan Keles[1], Sjoerd JB Holwerda[1], Hubertus Kohler[1], Filippo M Rijli[1,2], Michael Stadler[1,3]*, Botond Roska[1,4]*

[1]Friedrich Miescher Institute for Biomedical Research, Basel, Switzerland; [2]University of Basel, Basel, Switzerland; [3]Swiss Insitute of Bioinformatics, Basel, Switzerland; [4]Department of Ophthalmology, University of Basel, Basel, Switzerland

**Abstract** High-resolution daylight vision is mediated by cone photoreceptors. The molecular program responsible for the formation of their light sensor, the outer segment, is not well understood. We correlated daily changes in ultrastructure and gene expression in postmitotic mouse cones, between birth and eye opening, using serial block-face electron microscopy (EM) and RNA sequencing. Outer segments appeared rapidly at postnatal day six and their appearance coincided with a switch in gene expression. The switch affected over 14% of all expressed genes. Genes that switched off were rich in transcription factors and neurogenic genes. Those that switched on contained genes relevant for cone function. Chromatin rearrangements in enhancer regions occurred before the switch was completed, but not after. We provide a resource comprised of correlated EM, RNAseq, and ATACseq data, showing that the growth of a key compartment of a postmitotic cell involves an extensive switch in gene expression and chromatin accessibility.

DOI: https://doi.org/10.7554/eLife.31437.001

**\*For correspondence:**
michael.stadler@fmi.ch (MS);
botond.roska@fmi.ch (BR)

**Competing interests:** The authors declare that no competing interests exist.

## Introduction

The nervous system extracts information from the environment via specialized sensory cells, which convert changes in physical quantities such as the number of photons, mechanical pressure, or the concentration of chemicals into neuronal signals. This conversion takes place in primary cilium-derived, dedicated neuronal compartments (*Avidor-Reiss et al., 2004*). Photons are detected in the outer segments of photoreceptors in the visual system. Mechanical pressure is detected in the various protrusions of mechanoreceptors in the somatosensory system and in the hair bundles of hair cells in the auditory and vestibular system. Concentration of chemicals is detected in the cilia of the olfactory receptor neurons in the olfactory system. These different, antenna-like, modified cilia are especially sensitive to genetic perturbations and are the most frequent sites of sensory loss (*Mitchison and Valente, 2017*; *Wheway et al., 2014*).

Image-forming vision is initiated by two types of photoreceptors: rods and cones. Human visual function is particularly dependent on cones, because they function during daylight and are the photoreceptors required for high-resolution vision and color vision (*Sahel and Roska, 2013*). Loss of the cone outer segment is the consequence of a number of genetic diseases, and causes blindness (*Sahel and Roska, 2013*). Once outer segments are lost, cones slowly degenerate. However, there is a significant time window during which degenerating cones have lost their outer segments (and hence their ability to detect light), but the rest of the cell remains alive (*Sahel and Roska, 2013*). Therefore, there is a therapeutic time window during which outer segments could potentially be

regenerated to restore light sensitivity. Understanding the molecular program that drives outer segment formation could provide insights into how to regenerate them.

There are two phases of specification from a progenitor cell to a functional cone photoreceptor during development: an early phase that determines the cellular fate; and a later phase that determines the full functionality of a cone. Cone fate is established in newly postmitotic progeny that are born from mitotic progenitor cells during embryonic development. In the mouse, this is between embryonic day 10 (E10) and E18 (*Brzezinski and Reh, 2015*; *Swaroop et al., 2010*; *Wang and Cepko, 2016*). Thus, the genetic fate of cones is established before birth. However, the key functional compartment, the outer segment, only develops postnatally (*Duncan and Herald, 1974*; *Obata and Usukura, 1992*; *Olney, 1968*; *Sedmak and Wolfrum, 2011*), and the cone can respond to light by the second postnatal week (*Gibson et al., 2013*). As cones are not dividing anymore at that stage, any regulation guiding the functional maturation needs to be implemented independently of the cell cycle. While much is known about the molecular program that establishes cone fate (*Emerson et al., 2013*; *Swaroop et al., 2010*), little is known about the molecular program that governs the formation of the cone outer segment.

To get an insight, we correlated the formation of outer segments, gene expression, and chromatin accessibility of mouse cones daily between birth and eye opening, using serial block-face electron microscopy, RNA sequencing, ATAC sequencing, and bioinformatic analyses. Our analysis concentrated on the central retina because the retina develops asynchronously, from the center to the periphery.

## Results

### Cone outer segments of the central retina appear rapidly at P6

We monitored the formation of outer segments in developing postmitotic photoreceptors in the central region of the mouse retina. We analyzed ten different time points from postnatal day 0 (P0) to P11 by 3D reconstructing the ultrastructure of the photoreceptor layer using serial block-face electron microscopy (*Busskamp et al., 2014a*; *Denk and Horstmann, 2004*). To ensure an intact ultrastructure of the outer segments we kept the photoreceptor layer together with the retinal pigment epithelium. We identified photoreceptor outer segments as subcellular structures with stacked electron-dense membranes at the tip of a connecting cilium. Outer segments were absent until P4, their number abruptly increased from 7% of the cells having an outer segment at P5 to 53% at P6, and then continued to increase until P11 (*Figure 1*, *Figure 1—figure supplement 1*).

With electron microscopy it is possible to definitely identify outer segments but it is not possible to differentiate between rods and cones at early postnatal stages. Given that the vast majority of mouse photoreceptors are rods (i.e. 97%) (*Carter-Dawson and LaVail, 1979*) our electron microscopy reconstruction is dominated by rod outer segments. To follow the development of cone outer segments specifically, but at lower resolution, we used a mouse line (Chrnb4-GFP) that selectively expresses GFP in cones (*Siegert et al., 2009*) and we monitored cone outer segment development using light microscopy. First, we verified the specificity of GFP expression in cones from P0 to P12 by immunohistochemistry using a mixture of antibodies against short- and middle-wavelength cone opsins. Between P3 and P11, 98% of GFP-positive cells were found above the inner plexiform layer (IPL, in the adult retina all photoreceptors are found above the IPL). 94% of those GFP-labeled cells were positive for cone opsin, indicating that they are cones (*Figure 1—figure supplement 2*). Conversely, 99.1% of opsin positive cells were labeled with GFP, indicating that most cones are GFP-labeled (*Figure 1—figure supplement 2*). Therefore, from P3 onwards, cones were specifically and extensively labeled in the Chrnb4-GFP mouse line. We then followed the distal tip of GFP-labeled cells every day from P0 to P12. Up to P4, opsin labeling was weak and it was confined to the cell body and inner segment, with no opsin-rich extension at the distal tip (*Figure 1B–C*). At P5, very few short opsin-labeled tip extensions appeared. We interpreted these extensions as outer segments. The number of opsin-rich extensions abruptly increased from 4% at P5 to 45% at P6, and then continued to increase until P12 (*Figure 1D*).

Therefore, both the electron and light microscopic observations suggest that the fraction of cones that has outer segments rapidly increased from P5 to P6 in the central retina (*Figure 1D*).

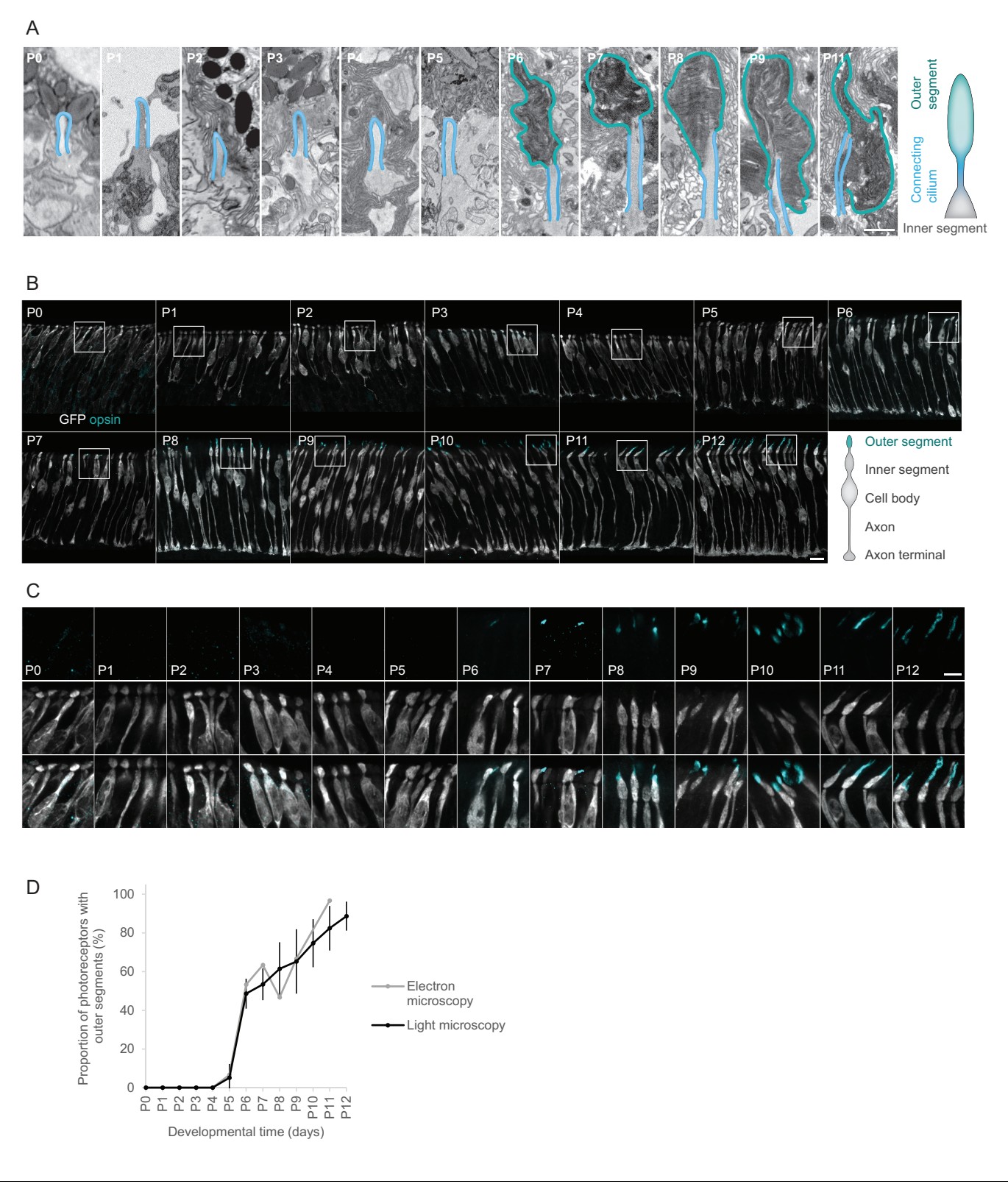

**Figure 1.** Outer segments of photoreceptors appear at P6. (**A**) Ultrastructure of photoreceptor compartments (outer segments in cyan; connecting cilium in blue) at different postnatal days. Each panel shows a single image from an image stack obtained by serial block-face electron microscopy, magnified from a region marked in **Figure 1—figure supplement 1**. Scale bar 1 µm. (**B**) Opsin expression in the retinas of Chrnb4-GFP mice at different postnatal days. Each image shows a maximum intensity projection of a 7 µm stack of confocal microscopy images obtained from a retinal

*Figure 1 continued on next page*

*Figure 1 continued*

vibratome section. Cones stained with antibodies against GFP (grey) and a mix of antibodies against s- and m/l-opsin (cyan). Scale bar 10 µm. (**C**) Magnification of regions marked in (**B**). Upper row: opsin staining. Middle row: GFP staining. Lower row: GPF and opsin staining. Scale bar 5 µm. (**D**) Quantification of outer segments shown in (**A**) (n = 1 mouse per day, 1 eye per mouse) and (**C**) (n = 3 mice per day, 1 eye per mouse). Error bars show means ± s.d.

DOI: https://doi.org/10.7554/eLife.31437.002

The following figure supplements are available for figure 1:

**Figure supplement 1.** The ultrastructure of photoreceptors at different postnatal days.
DOI: https://doi.org/10.7554/eLife.31437.003

**Figure supplement 2.** The identity of GFP-labeled cells in the retina of Chrnb4-GFP mice.
DOI: https://doi.org/10.7554/eLife.31437.004

## Genome-wide gene expression switch at P6

To correlate the appearance of cone outer segments with the gene expression patterns of developing cones, we determined the transcriptomes of postmitotic cone photoreceptors in the central region of the mouse retina every day between birth (P0) and eye opening (P12). At each postnatal day we isolated GFP-labeled cells from three different Chrnb4-GFP mice (biological triplicates) using fluorescence-activated cell sorting. We then acquired the transcriptomes of the sorted cones using next generation RNA sequencing. Our data set contained 39 transcriptomes (*Figure 2—figure supplement 2*).

The transcriptomes of cones isolated before P6 correlated strongly with each other (mean Pearson R = 0.96) (*Figure 2A*). Likewise, correlation was high among transcriptomes of cones isolated after P6 (mean Pearson R = 0.97). However, the correlation between cones isolated before P6 and cones isolated after P6 was significantly lower (mean Pearson R = 0.8), suggesting a switch in gene expression around P6.

This switch could be one of many different temporal changes in gene expression between P0 and P12. Other types of temporal changes include the continuous rise of expression of some genes, and the continuous decrease of expression of others. To rank the different types of changes, we performed a principal component analysis (Materials and methods). Each principal component describes a commonly occurring temporal change in gene expression, and its weight shows how much of the variance in the expression of all genes can be explained by this time course. The first principal component was characterized by a switch at P6: similar values from P0 to P5, a sudden change at P6, and then again similar values from P7 to P12 (*Figure 2B*). This principal component accounted for 67% of the variance in gene expression, while the second principal component was responsible for only 5% of the variance (*Figure 2C*). Therefore, a switch in gene expression at P6 is the dominant event in the time course of postnatal gene expression in the first two postnatal weeks.

A switch could mean a sudden decrease or an increase in gene expression. To understand the polarity of the change, we sorted the expressed genes according to the fold change of expression before and after P6. We found a group of genes that switched on (n = 508 'switch-on' genes) and another group of genes that switched off (n = 1038 'switch-off' genes) between 2- and 1,000-fold (likelihood-ratio test, false discovery rate (FDR) < 5%, *Figure 2D–E*). Therefore, 14% of all the 11,257 expressed genes switched expression at least 2-fold around P6 ('switch genes').

To determine whether the switch was a genome-wide phenomenon or whether particular chromosomes were more affected than others we analyzed the distribution of switch genes in the genome. We found that switch genes were distributed over the whole genome. The distribution of switch-on and switch-off genes was not different from the distribution of all genes (Chi square test, switch-on: p=0.279, switch-off: p=0.5499) (*Figure 2—figure supplement 1*). We then analyzed the finer time course of switch-on and switch-off genes that switched at least 4-fold (n = 122 'switch-on' genes and n = 488 'switch-off' genes, likelihood-ratio test, FDR < 5%). We found that the switch started at P5 and ended at P7 (*Figure 2F–H*). Thus, the switching genes are up- or downregulated at P6.

The switch in gene expression coincided with the sudden appearance of the outer segments: on the same day that the gene expression switch began, the growth of outer segments started (*Figure 2I*).

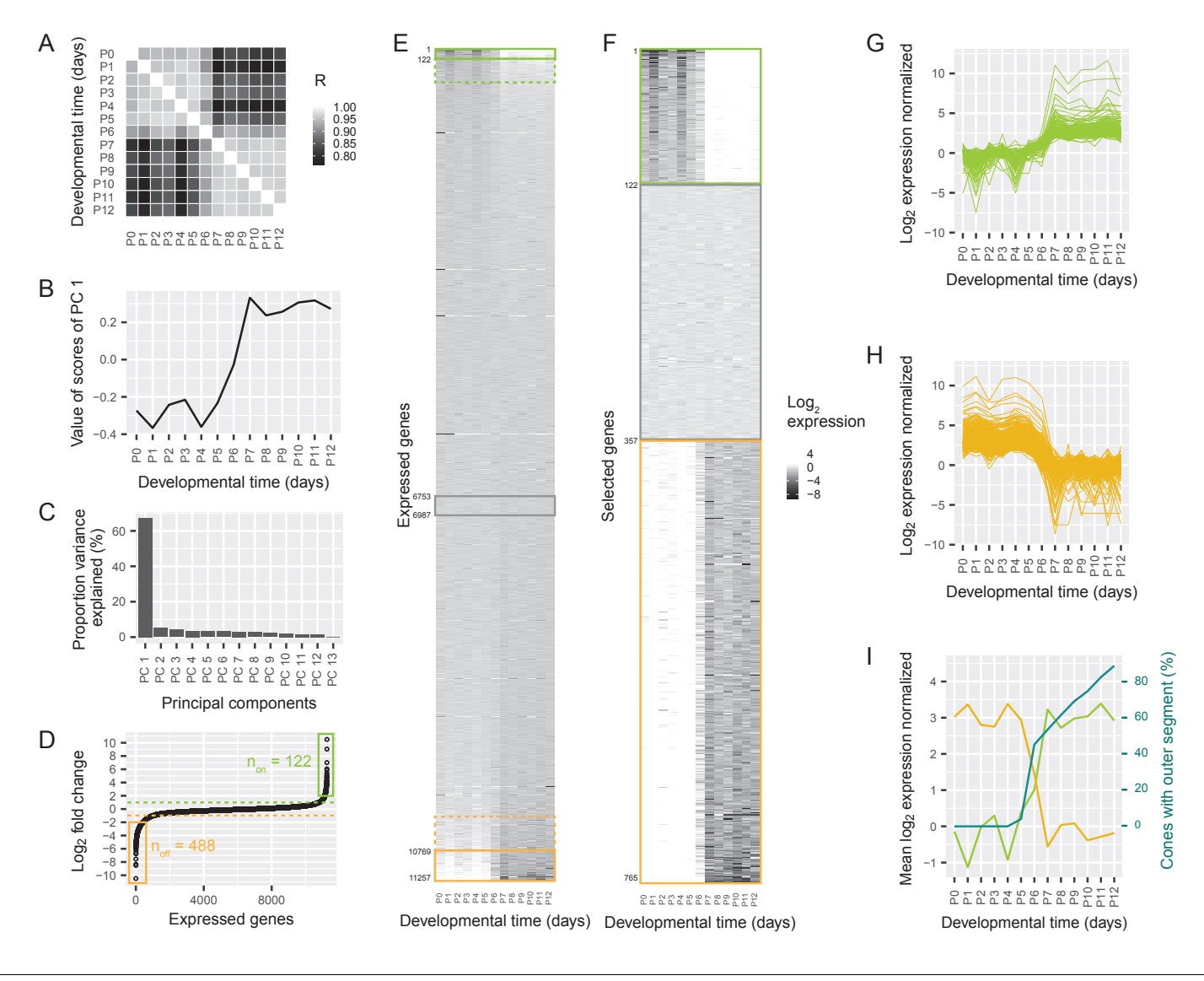

**Figure 2.** Switch in gene expression in cone photoreceptors at P6. (**A**) Pearson correlations between the $\log_2$-transformed numbers of RNAseq reads per gene in cones isolated at different postnatal days from Chrnb4-GFP mice, three replicates per time point. R = Pearson correlation coefficient. (**B**) The values of the first principal component, representing a common time course. (**C**) The proportion of variance explained by different principal components. (**D**) Ordered $\log_2$ fold changes in expression of all 11,257 expressed genes in isolated cones. Fold changes are the difference in expression levels before (mean expression P0-P5) and after (mean expression P7-P12) the switch. Fold changes are ordered from smallest to largest value. Switch-on ($n_{on}$) and switch-off ($n_{off}$) genes with absolute $\log_2$ fold changes higher than 2 (= $\log_2$ fold changes higher than 2 or lower than −2) are marked with a box. Switch-on and switch-off genes with absolute $\log_2$ fold changes higher than 1 are above or below the dashed lines. (**E–F**) Heat maps showing expression levels for all expressed genes (**E**) and selected genes (**F**) from P0-P12. Rows show genes ranked by $\log_2$ fold change. Green and orange boxes correspond to regions marked in (**D**). The grey box marks constant genes with absolute $\log_2$ fold changes less than 0.01. (**E**) Expression levels of all expressed genes. (**F**). Expression levels of switch-on and switch-off genes that have absolute $\log_2$ fold changes greater than 2, as well as constant genes. (**G–H**) Time course of normalized $\log_2$ fold changes for switch-on genes (**G**) and switch-off genes (**H**) that have absolute $\log_2$ fold changes greater than 2. The data are normalized to the mean expression before the switch (P0–P5) (**G**) or after the switch (P7–P12) (**H**). (**I**) Mean $\log_2$ fold changes of switch-on and switch-off genes with absolute $\log_2$ fold changes greater than 2 and percentage of outer segments (replotted from *Figure 1D*).

DOI: https://doi.org/10.7554/eLife.31437.005

The following figure supplements are available for figure 2:

**Figure supplement 1.** Switch genes are evenly distributed over the genome.

DOI: https://doi.org/10.7554/eLife.31437.006

**Figure supplement 2.** RNAseq quality control.

*Figure 2 continued on next page*

*Figure 2 continued*

DOI: https://doi.org/10.7554/eLife.31437.007

## The identity of switch-on and switch-off genes

To better understand the gene expression switch, we analyzed which genes, gene classes, and pathways were involved. A large-scale transition in gene expression likely involves transcription factors. We therefore analyzed the frequency and identity of transcription factors among switch genes. Among switch-off genes, we found four times more transcription factors than expected by chance: 23% of the top 100 switch-off genes were transcription factors, whereas the expected frequency of transcription factors in the entire transcriptome was 6% (Permutation test, $p=9 \times 10^{-8}$) (*Figure 3A*). In comparison, the frequency of transcription factors among the top 100 switch-on genes was not significantly different from the expected frequency (Permutation test, p=0.21). The switch-off genes included five neurogenic transcription factors that are known to promote direct conversion of cultured fibroblasts or induced pluripotent stem cells to neurons (*Busskamp et al., 2014b*; *Pfisterer et al., 2011*; *Son et al., 2011*; *Vierbuchen et al., 2010*; *Zhang et al., 2013*). All five of these genes were strongly downregulated: *Ascl1* 188-fold, *Myt1l* 13-fold, *Neurog1* 74-fold, *Neurog2* 21-fold, and *Pou3f2* (also known as *Brn2*) 7-fold. Note that the expression of *Neurog1* and *Pou3f2* was low also before the switch. A list of the transcription factors that switch off, is shown in *Supplementary file 1b*.

Many cone-specific proteins, including members of the phototransduction cascade, are located in the outer segment. As the growth of the outer segments and the regulation of switch genes began in parallel, we asked how cone-specific genes were distributed among switch genes. We defined 41 cone-specific genes based on a cell type transcriptome study (*Siegert et al., 2012*) (*Supplementary file 1a*). The expression of this group of genes switched on significantly (Permutation test, $p<10^{-6}$) (*Figure 3B*, *Supplementary file 1a*). Likewise, the phototransduction pathway switched on significantly (Permutation test, $p<10^{-6}$) (*Figure 3B*). Therefore, cone-specific genes and genes of the phototransduction pathway were among the switch-on genes.

Performing a global analysis for 219 other cellular pathways revealed that 50 pathways were significantly up- or downregulated in a switch-like manner (Permutation test, p<0.05) (*Figure 3—figure supplement 1* and *Supplementary file 1c*). Among those 50 switch pathways, all of the metabolic pathways switched on, which shows a boost in energy metabolism at the time of outer segment formation (*Figure 3B*). On the other hand, pathways involved in Hedgehog signaling, Notch signal transduction, and axon guidance switched off (Permutation test, p=0.005, p=0.01, and $p<10^{-6}$, respectively) (*Figure 3B*).

Therefore, pathways essential for establishing a functional cone – with phototransduction and a high energy metabolism – switched on. In contrast, components of general neuronal development – such as neurogenic transcription factors and the axon guidance pathway – switched off.

## Regulation of the switching genes

A switch in gene expression can be regulated transcriptionally or post-transcriptionally. To differentiate between these two scenarios, we performed an exon-intron split analysis (*Gaidatzis et al., 2015a*). We separately quantified RNA sequencing reads obtained from exons (from both pre-mRNA and mature mRNA) and reads obtained from introns (only from pre-mRNA), and we correlated the change of exonic and intronic reads before and after the switch. If a transcriptional mechanism is responsible for the switch, then the observed changes in exonic and intronic reads are expected to correlate. If, on the other hand, a post-transcriptional mechanism controls the switch, exonic and intronic reads are decoupled and a lower correlation is expected. The Pearson correlation between the changes in exonic and intronic reads was 0.84, suggesting that transcriptional changes accounted for at least 70% of the changes in gene expression during the switch (*Figure 4A*).

As the switch was mostly regulated by a transcriptional mechanism, chromatin rearrangements could be involved in the regulation. To understand if chromatin architecture was rearranged, we mapped the accessible genome by ATACseq. This allowed us to determine which regions of the genome were free of nucleosomes and thus accessible to protein binding. We performed ATACseq

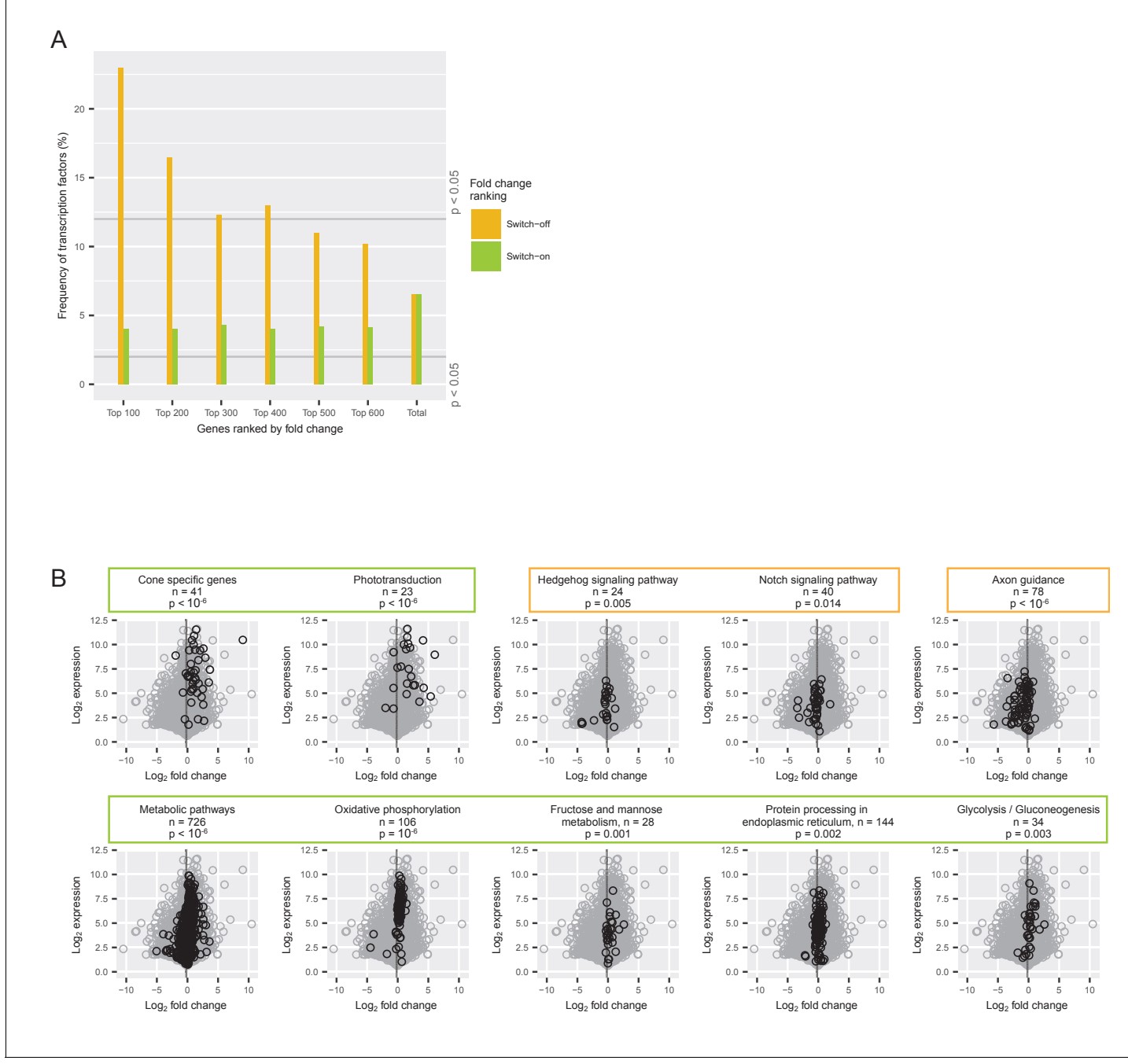

**Figure 3.** Gene classes and pathways involved in the switch. (**A**) Proportion of transcription factors among different groups of switch-on (green) and switch-off (orange) genes. All expressed genes were ordered by their $\log_2$ fold changes between P0-P5 and P7-P12 and the 100, 200, 300, 400, 500 and 600 most strongly on- or off-switching genes were grouped. The last group 'total' contains all expressed genes (for conceptual consistency there are two separate bars but there is no difference between the orange and the green bar). The grey lines mark the frequency of transcription factors that is significantly higher or lower than expected (permutation analysis). (**B**) Comparison of $\log_2$ fold changes and $\log_2$ maximum expression levels during P0-P12 of all expressed genes (grey) with genes associated with a particular pathway (black): pathways needed for cone function, signaling pathways, axon guidance pathway, and metabolic pathways. The grey line marks the mean fold change of all expressed genes (−0.18). P-values from permutation analyses; n is the number of genes associated with each pathway. Orange boxes mark pathways that switch off, and green boxes mark pathways that switch on.

DOI: https://doi.org/10.7554/eLife.31437.008

The following figure supplement is available for figure 3:

**Figure supplement 1.** Gene classes and pathways involved in the switch.

*Figure 3 continued on next page*

*Figure 3 continued*

DOI: https://doi.org/10.7554/eLife.31437.009

at three time points: before the switch (P3), on the day of the switch (P6), and after the switch (P10). We obtained 70% mappable sequences (*Figure 4—figure supplement 1*), and identified 203,285 peaks of accessible DNA regions (regions of open chromatin). Regions of open chromatin correlated between P6 and P10, while the correlation between P3 and P6 was lower (*Figure 4B*). We then counted regions that either lost or gained accessibility between P3-P6 and P6-P10, and we found a striking asymmetry (*Figure 4C*). The vast majority of the changes happened before the transcriptional switch was completed (between P3 and P6), when 26,014 regions lost accessibility and 5687 regions gained accessibility. In contrast, only 172 regions lost and 114 gained accessibility after the switch (between P6 and P10).

We next compared chromatin accessibility of P3, P6, and P10 cones to adult cones and adult rods (*Mo et al., 2016*). As expected, all three samples correlated strongly with adult cones (R = 0.86), whereas, they correlated much less with adult rods (R = 0.32) (*Figure 4—figure supplement 1*). Furthermore, P6 and P10 cones correlated more with adult cones (R = 0.88) than P3 cones (R = 0.81).

The chromatin changes observed between P3 and P6 showed two characteristics. First, short regions changed more than large regions (*Figure 4—figure supplement 1*), and, second, distal regions changed more than proximal regions (*Figure 4D*). Proximal regions are promoters, which lie within several hundred base pairs of the nearest transcription start site and are usually longer accessible regions. Distal regions are typically enhancers, which lie more than 1 kb away from the nearest transcription start site and are generally short accessible regions. This suggests that there is a global chromatin accessibility loss that affected distal enhancers but not promoters.

A change in chromatin accessibility is often caused by an altered interaction between DNA binding proteins and specific regions on the DNA. To identify potentially relevant DNA binding proteins, we searched for distinct sequence motifs in differential peaks between P3 and P6. We used a regularized regression model (*Friedman et al., 2010*) to analyze how the presence of known DNA binding motifs contributed to the observed chromatin changes (*Figure 4—figure supplement 1*). Currently, only 23% of the transcription factors that are expressed in cones before or after the switch have annotated binding sites, which limits a full analysis. Nonetheless, we identified a single DNA binding protein, CTCF (*Ong and Corces, 2014*; *Phillips and Corces, 2009*), with an outstanding contribution to explaining the observed accessibility changes (*Figure 4—figure supplement 1*). Are regions that can bind CTCF protected from chromatin accessibility changes before the transcriptional switch? We quantified the number of CTCF binding sites within each region of open chromatin, and found that the more CTCF binding sites a region contained, the more stable its chromatin accessibility remained (*Figure 4E*). This suggests that regions without CTCF binding sites had a higher probability of losing accessibility between P3 and P6, than regions with CTCF binding sites.

Therefore, the switch was mostly regulated transcriptionally, and genome-wide chromatin rearrangements accompanied the switch in an asymmetric fashion: (1) they occurred before the switch was completed (P3 vs P6), but not after, and (2) mostly the chromatin conformation closed, but not opened. These chromatin changes predominantly affected enhancers, and in particular regions that lacked CTCF binding sites.

## Discussion

In this work we created a resource of correlated EM, RNAseq, and ATACseq data, which allowed us to correlate ultrastructure and gene expression of cones during postmitotic development. We found a major switch in cone morphology and gene expression at P6 (*Figure 5*). The switch involves 14% of all expressed genes, and coincides with the formation of cone outer segments. Pathways essential for establishing a functional cone switch on, while components of general neuronal development switch off. The switch is mostly regulated transcriptionally, and is accompanied by a genome-wide loss in chromatin accessibility. Enhancer elements – particularly regions without CTCF binding sites - were mostly affected by the chromatin accessibility loss. To our knowledge, no such fast and

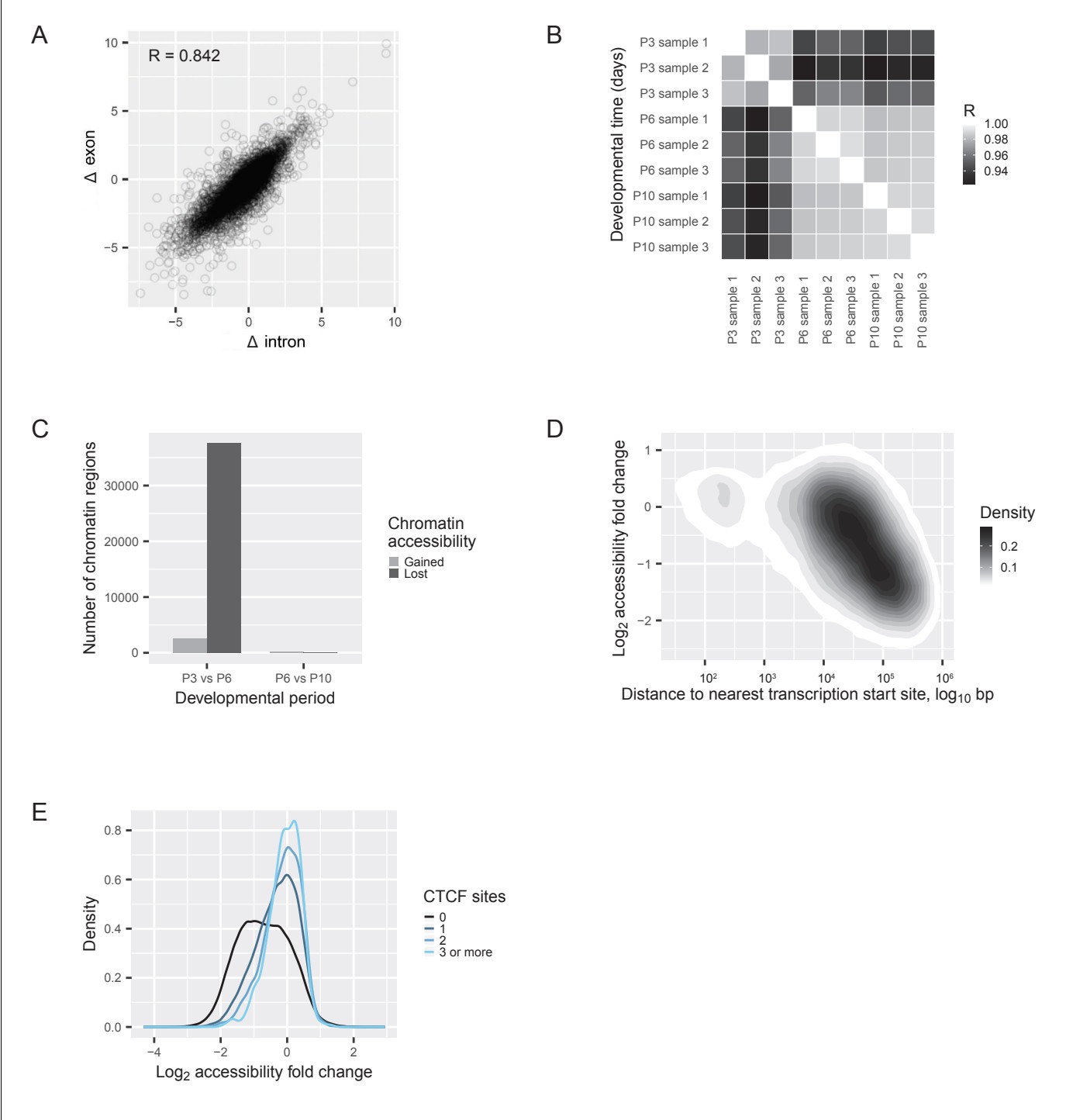

**Figure 4.** Transcriptional regulation of the switch. (**A**) RNAseq reads quantified with an exon-intron split analysis (*Gaidatzis et al., 2015a*) to differentiate transcriptional and post-transcriptional regulation of the switch. Correlation of $\log_2$ fold changes ($\Delta$) in reads obtained from exons (from both pre-mRNA and mature mRNA) with reads obtained from introns (only from pre-mRNA). R = Pearson correlation coefficient. (**B**) ATACseq samples: Pearson correlations between the numbers of reads at each peak (a 'peak' is a region of accessible chromatin) across different developmental days. R = Pearson correlation coefficient. (**C**) Changes of chromatin accessibility before (P3 vs. P6) and after (P6 vs. P10) the switch. The number of chromatin regions gaining (gray) or losing (black) accessibility is shown. The threshold was an absolute $\log_2$ fold change of more than one and a false discovery rate less than 0.01. (**D**) Changes of chromatin accessibility in identified peaks between P3 and P6 depending on the distance to the nearest transcription start site. The density of regions of accessible chromatin as a function of distance and $\log_2$ fold change of accessibility is shown. Density is indicated according to the gray scale on the right. (**E**) Changes of chromatin accessibility between P3 and P6 depending on the presence of CTCF binding sites.
*Figure 4 continued on next page*

*Figure 4 continued*

The number of binding sites for the protein CTCF was predicted for each region of accessible chromatin, and the regions of accessible chromatin were separated into four groups: regions without CTCF binding sites (black); regions containing 1 binding site (dark blue); regions containing 2 binding sites (blue); and regions containing 3 or more binding sites (light blue). The density of regions of accessible chromatin was plotted as a function of the $\log_2$ fold changes of chromatin accessibility, for each group separately.

DOI: https://doi.org/10.7554/eLife.31437.010

The following figure supplement is available for figure 4:

**Figure supplement 1.** ATACseq quality control and modelling.

DOI: https://doi.org/10.7554/eLife.31437.011

extensive change in gene expression and chromatin accessibility, correlated with a distinct morphological change, has been reported in neurons.

Our structural analyses indicate that the formation of cone outer segments starts between P5 and P6 in the central retina. Outer segment formation is rapid and synchronized, and by P6, 53% of the outer segments are already present in some form. It is known that the retina develops in an asynchronous fashion: the central part of the retina being at an advanced developmental stage compared to the periphery (*Holt et al., 1988*). This wave of maturation explains why previous studies analyzing the whole retina found a more gradual change compared to what we found analyzing only the central retina (*Duncan and Herald, 1974*; *Obata and Usukura, 1992*; *Olney, 1968*; *Sedmak and Wolfrum, 2011*).

Do rod outer segments also appear rapidly at P6? Our electron microscopy reconstructions of the photoreceptor layer contained both cone and rod photoreceptors. Morphologically, cones and rods are indistinguishable during early postnatal development. Since rods outnumber cones in the mouse retina, rod outer segments must also be present in our analysis and, therefore, also appear rapidly at P6. In this work, we chose to investigate cones, for two reasons. First, because they are highly important for human daylight vision. Second, because the determination of the cone cell fate occurs prenatally, which is shifted in time from outer segment formation. This shift allows the clear separation of the changes in gene expression associated with the two processes. Based on the structural and functional similarities between rods and cones, and based on a recent study, which showed

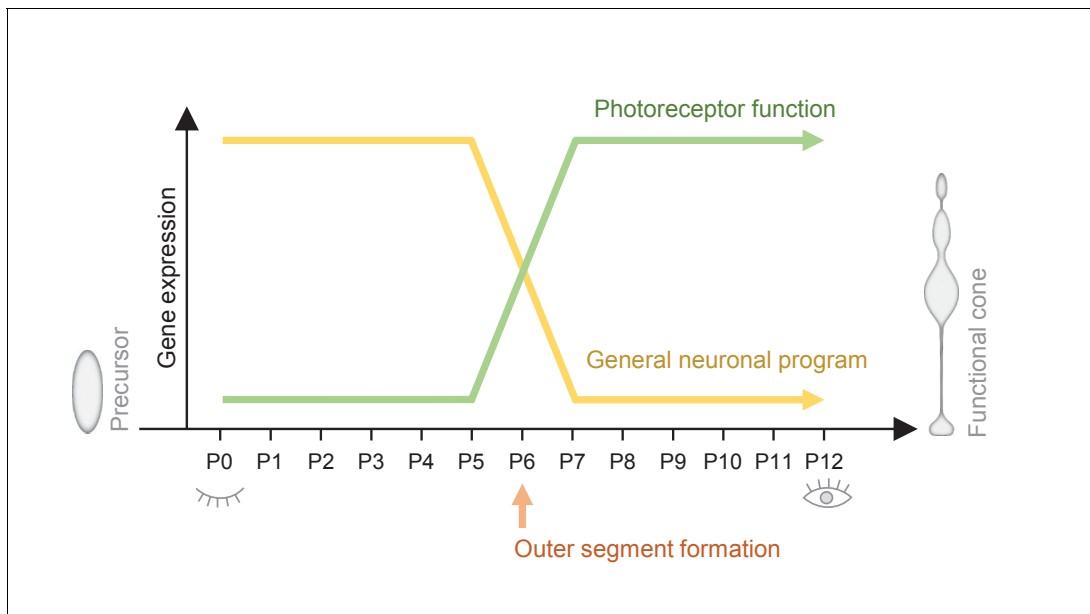

**Figure 5.** Summary of the switch that happens during maturation from a cone precursor to a functional cone. Changes in cone morphology and gene expression are synchronized and locked to P6. At P6, outer segments form and pathways essential for establishing a functional cone switch on, while components of general neuronal development switch off.

DOI: https://doi.org/10.7554/eLife.31437.012

large-scale gene expression changes between P6 and P10 in rod photoreceptors (*Kim et al., 2016*) as well as an earlier study showing similar changes using SAGE and in situ hybridization (*Blackshaw et al., 2001*; *Blackshaw et al., 2004*), it is possible that rods also show a gene expression switch between P5 and P7. Further studies at a fine temporal resolution are needed to investigate this possibility.

The analysis of gene classes and pathways that are involved in the switch has revealed insights into the function of the switch. First, switch-off genes contain a high proportion of transcription factors, including *Ascl1*, *Myt1l*, *Pou3f2* (also known as *Brn2*), *Neurog1*, and *Neurog2*. These neurogenic transcription factors are known to be - in different combinations - necessary and sufficient to promote direct conversion of cultured fibroblasts or induced pluripotent stem cells to neurons (*Busskamp et al., 2014b*; *Pfisterer et al., 2011*; *Son et al., 2011*; *Vierbuchen et al., 2010*; *Zhang et al., 2013*). In neurons, these factors continue being expressed throughout the lifetime of the cell (*Atlas, 2016*). This suggests that the neuronal state is turned off before cones become functional sensory cells. A potential reason could be that these proneuronal factors play a role in axon formation, and the same factors may inhibit the formation of outer segments. The downregulation of these factors may initiate the outer segment formation. Second, switch-on genes include pathways needed for light detection (*Siegert et al., 2012*), as well as pathways for energy supply necessary for light detection. The late switch-on of metabolic pathways may save energy until it is needed for maintaining light responses.

The mechanism of the switch involves epigenetic changes. We found that the switch is mostly controlled transcriptionally, which we determined using an exon-intron split analysis. This is consistent with our finding that the switch was accompanied by a large-scale loss of chromatin accessibility. This loss of chromatin accessibility only affected regions distal to the transcriptional start site, suggesting that the switch decreases accessibility at enhancer elements. Interestingly, the chromatin changes preferentially targeted regions that lacked CTCF binding sites. CTCF participates in the global organization of chromatin architecture by regulating the interplay between higher-order chromatin structure and cell-type-specific gene expression during development (*Ong and Corces, 2014*; *Phillips and Corces, 2009*). These findings indicate that a global chromatin rearrangement occurs during the switch.

What triggers the switch in gene expression at P6? Either a precisely timed regulatory event from outside the retina, or an internal clock within the retina. An internal clock would be the accumulation of a signal that triggers the switch once it reaches a threshold. This clock could reside either in cones or in other retinal cells. *Opn1mw* (encoding middle-wavelength opsin) is one of the genes that is not expressed in cones before P6 and whose expression switched on at P6. *Opn1mw* expression also turns on in vitro in cultured retinal explants (*Satoh et al., 2009*; *Wikler et al., 1996*), if the retina is cultured from P3 onwards. If the other switch genes behave similarly to *Opn1mw* in vitro, the switch is either triggered by an intra-retinal clock or by an extra-retinal signal that acts before P3 and has a delayed effect. Intriguingly, there is another switch-like event at P6 in another type of postmitotic retinal neuron, the retinal ganglion cells. In retinal ganglion cells, the neurotransmitter GABA is excitatory before P6 and inhibitory after P6. This excitatory-to-inhibitory switch is caused by a sudden change in chloride concentration within ganglion cells (*Zhang et al., 2006*). For both the excitatory-to-inhibitory switch in ganglion cells (*Barkis et al., 2010*) and the gene expression switch in cones that we report here, the trigger is not known. Elucidating the trigger for the gene expression switch in cones could help to establish a causal relationship between the switch and the formation of outer segments, and may help to find strategies to regenerate the outer segments of cones in blinding diseases.

Are miRNAs involved in the switch? The sensory-cell-specific miRNA-182/183/96 cluster was implicated in outer segment growth. First, this cluster was shown to be necessary for outer segment maintenance in adult cone photoreceptors (*Busskamp et al., 2014a*). Second, it was shown that the expression of the cluster starts at P10 (*Krol et al., 2015*) and, third, that knocking-out of these miRNAs results in shortened outer segments (*Fan et al., 2017*; *Xiang et al., 2017*). Taken together, these studies suggest that this miRNA cluster is not involved in initiating the formation of outer segments at P6, but that it is needed later in development, for maintenance or perhaps also for elongation of outer segments.

## Materials and methods

### Data

EM data are deposited on Dropbox, at https://www.dropbox.com/sh/48euwi0gno07j7s/AACALr_ZMVDF9d20SrYa8yC7a?dl=0. No password is required.

Sequencing data are deposited on GEO with the SuperSeries accession number GSE97536. The SubSeries accession numbers are GSE97534 for RNAseq and GSE97535 for ATACseq.

### Animals

Tg(Chrnb4-EGFP) mice were purchased from MMRRC. Animal procedures were done in accordance with standard ethical guidelines (European Communities Guidelines on the Care and Use of Laboratory Animals, 86/609/EEC) and were approved by the Veterinary Department of the Canton of Basel-Stadt. Mice were housed with a 12 hr dark-light cycle.

### Immunohistochemistry

After eyes were enucleated, they were either fixed as a whole eye for 3 hr or retinas were dissected and only the retinas were fixed for 30 min. The fixative was 4% paraformaldehyde (Sigma-Aldrich, St. Louis, Missouri) in phosphate-buffered saline (PBS). If the whole eye was fixed, eyecups were dissected after fixation by removing the anterior chamber of the eye in PBS. Eyecups and retinas were washed in PBS three times for a minimum of 30 min at room temperature (RT). The samples were dehydrated in 30% sucrose for minimum 30 min at RT and then treated with three freeze-thaw cycles. All eyecups and some of the retinas were embedded in 3% agarose (SeaKem Le Agarose, Lonza, Switzerland) in PBS, and 150 µm sections were cut using a Leica VT1000S vibratome. The remaining retinas were processed without sectioning. Sections and whole-mount retinas were incubated in blocking solution for 1 hr at RT. The blocking solution contained 10% normal donkey serum (Chemicon, Waltham, Massachusetts), 1% bovine serum albumin, and 0.5% Triton X-100 in PBS, at pH 7.4. Antibodies were diluted in 3% normal donkey serum, 1% bovine serum albumin, 0.02% sodium acid, and 0.5% Triton X-100 in PBS. Primary antibodies were applied for at least 1 day. The following antibodies were used: chicken-anti-GFP (1:500, Chemicon, AB16901), rat-anti-GFP (1:500, Nacalai/Brunschwig, Japan, 04404–84), goat-anti-S-opsin (1:200, Santa Cruz, Dallas, Texas, sc-14365), goat-anti-M/L-opsin (1:200, Santa Cruz, sc-22117), rabbit-anti-S-opsin (1:200, Millipore, Burlington, Massachusetts, AB5407), and rabbit-anti-M/L-opsin (1:200, Millipore, AB5405). Retinas were then washed three times for 10/30 min (sections/whole mounts) in PBS, and incubated in the following secondary antibodies, all diluted at 1:200: Alexa Fluor 488 Donkey Anti-Chicken (Jackson Immuno, Bar Harbor, Maine, 703-545-155), Alexa Fluor 488 Donkey Anti-Rat (Life Technologies, Carlsbad, California, a21208), Alexa Fluor 555 Donkey Anti-Rabbit (Thermo Fisher, Waltham, Massachusetts, A31572), Alexa Fluor 568 Donkey Anti-Goat (Thermo Fisher, A11057), Alexa Fluor 647 Donkey Anti-Rabbit (Life Technologies, A31573), and Alexa Fluor 647 Donkey Anti-Goat (Thermo Fisher, A21082). Nuclei were stained with 10 µg/ml of Hochst (Thermo Fisher, 62249) at a dilution of 1:500. Secondary antibodies and Hochst were applied for 1 hr at RT, followed by three washes in PBS of 10/30 min each (sections/whole mounts), both in the dark. Sections and whole mounts were mounted on slides with ProLong Gold antifade reagent (Thermo Fisher, P36934).

### Light microscopy

Confocal three-dimensional scans were taken with a Zeiss LSM 700 Axio Imager Z2 laser-scanning confocal microscope using Plan-Apochromat 63×/1.40 Oil DIC M27 or EC Plan-Neofluar 40×/1.30 Oil M27 oil-immersion objective lenses at four excitation laser lines (405 nm for Hochst, 488 nm for GFP, and 555 nm or 639 nm for opsin). All images were taken within retina's central disk of 3 mm in diameter (the center being the optic nerve). Optical sections of 1 µm were acquired.

### Image analysis light microscopy

Zeiss three-dimensional scan files (lsm format) were opened in Fiji ImageJ and processed as follows for quantification. First, optical sections were projected using a maximum intensity projection. For the compartmentalization analysis in vibratome sections, seven optical sections were projected. For the control experiments in vibratome sections, five optical sections were projected. For the control

experiments in whole mounts, a single plane at the level of the inner and outer segment was analyzed. If necessary, the minimum or maximum pixel values were narrowed. Brightness or contrast were never changed. Cells were counted manually using the Cell Counter plugin. For the compartmentalization analysis, all cells (22 cells on average) were counted in each image (n = 9 (3 animals, 3 regions each)), and the presence/absence of an outer segment was quantified. The example images in *Figure 1* are identical to the quantified images. For the control experiments in vibratome sections, all GFP-positive cells across all layers of the retina (39 cells on average) were counted in each image (n = 3 (3 animals, 1 region each)), and the presence of GFP-positive cells outside the photoreceptor layer was quantified. For the control experiments in whole mounts, all cells were counted (112 cells on average) in each image (n = 1), and it was quantified how many cells were positive for GFP and opsin (labeled cones), and how many were only positive for GFP (labeled rods) or only positive for opsin (unlabeled cones).

## Sample preparation electron microscopy

Eyes were enucleated and a small hole was made in the eye using a 27 G needle to allow influx of fixative. From the hole, the cornea was cut three times with small scissors towards the sclera, approximately 2 mm per cut, to further open the eye. Eyes were placed in fixative containing 2% paraformaldehyde (EMS, Hatfield, Pennsylvania, 15700), 1% glutaraldehyde (EMS 16300), and 2 mM $CaCl_2$ (Sigma Aldrich, St. Louis, Missouri, 53704) in 0.3 M cacodylate (Sigma Aldrich 20838) at pH 7.4. Eyes were fixed overnight at 4°C. The next day, retinas were dissected in 0.15 M cacodylate. Retinas were then vibratome sectioned as described above. The sections were stained using a standard protocol (https://www.ncmir.ucsd.edu/sbem-protocol/) with small modifications. Retinal sections were washed five times 3 min in 0.15 M cacodylate. Next, they were postfixed in a reduced osmium solution for 1 hr on ice. The solution contained 3% $K_4Fe(CN)_6$ (Sigma Aldrich, 31254), 0.3 M cacodylate buffer, and 4% aqueous $OsO_4$ (EMS, 1/9100). Then, the samples were washed five times 3 min in $ddH_2O$ and incubated in TCH solution for 20 min at RT. The TCH solution was made freshly each time, by incubating 1% thiocarbohydrazide (Sigma Aldrich, 88535) in $ddH_2O$ for 1 hr at 60°C. It was swirled three times during the incubation and cooled down to RT after. Before usage, it was filtered using a 0.22 µm syringe filter (VWR, Radnor, Pennsylvania, 514–0072). The second postfixation was done using 2% $OsO_4$ in $ddH_2O$ for 30 min at RT. This was followed by five times 3 min wash in $ddH_2O$. The samples were then incubated in 1% uranyl acetate in $ddH_2O$ overnight, and afterwards washed five times 3 min in $ddH_2O$. Next, they were incubated in Walton's lead aspartate for 20 min at 60°C. Walton's lead aspartate was prepared by adding 0.66% lead nitrate (EMS, 17900) to 60°C warm 0.03M aspartic acid (Sigma Aldrich, 1043819), and the mix was incubated at 60°C for 30 min. The pH was adjusted to 5.5 with 1 M NaOH (Sigma Aldrich, 72068) at 60°C. After Walton's lead aspartate treatment, retinal sections were washed five times 3 min in $ddH_2O$ at RT, and afterwards dehydrated with ethanol, each concentration for 2 min: 50%, 70%, 90%, 95%, 100%, and 100% plus an additional 2 min in cold propylene oxide. Then the samples were incubated in 50% propylene oxide and 50% resin for 2 hr. For the resin, 20 ml Embed 812 (Serva, Germany, 21045), 6.25 ml 2-dodecenylsuccinic acid anhydride (Serva, 20755), 5 ml methyl nadic anhydride (Serva, 29452), and 0.325 ml N-benzyldimethylamine (Serva, 14835) were mixed. After that, samples were immersed in 100% resin for 1 hr. Then, the sample was incubated in fresh resin overnight. Depending on the microscope used for imaging, samples were either embedded in fresh resin or in silver resin and dried at 60°C for 48 hr. Silver resin (Epo-Tek, Billerica, Massachusetts, EE129-4) was a mix of 1.25 g component A and 1 g component B. The embedded tissue blocks were taken from the retina's central disk of 3 mm in diameter (the center being the optic nerve), and trimmed to a size of 1000 × 300×1000 µm using a razor blade and glued on aluminum stubs for SBEM (Gatan, Pleasanton, California) using cyanoacrylate glue.

## Electron microscopy.

Retinas were imaged using scanning electron microscopes (Quanta, FEI, Hillsboro, Oregon, and Merlin, Zeiss, Germany). Inside each microscope was an ultramicrotome (3View, Gatan), which can remove thin tissue sections and enable the acquisition of serial images of the block face of the tissue. Backscattered electrons were detected and the 3View DigitalMicrograph software was used to

acquire the images. Images were acquired at pixel sizes of 2.1 to 16.7 nm and a section thicknesses of 50 to 100 nm.

## Image processing

The serial images in dm3 format were imported into Fiji ImageJ. They were aligned using the automated alignment tool (multilayer mosaic) in TrackEM (*Cardona et al., 2012*), allowing only for translational movements. Compartments were counted manually in TrackEM. In each image stack, 30 cells were analyzed for the presence or absence of an outer segment. Outer segments were identified as stacked electron-dense membranes at the tip of a connecting cilium.

## RNA sequencing sample preparation

A protocol from Siegert et al. (*Siegert et al., 2012*) was used to dissociate retinal cells. Retinas were dissected in HBSS and the retina's central disk of 3 mm in diameter (the center being the optic nerve) was isolated with scissors. The central retina was incubated in an activated papain solution for 5 min at 37°C. After removing the papain solution, retinas were washed with HBSS containing 2% FCS and then dissociated in HBSS containing 2% by triturating the tissue with a fire-polished Pasteur pipette. The suspension was filtered before FACS sorting. Per time point (P0 till P12), we FACS-sorted three retinas, each from a different animal. If 20,000 cells could not be obtained from one retina, retinas of different animals were pooled. Cells where sorted using a BD Influx cell sorter (Becton, Dickinson and Company, BD Biosciences, San Jose, CA USA) with the BD FACS Sortware sorter software (Version 1.01.654). The sort was performed with a 100 μm nozzle tip, at a sheath pressure of 19.0 psi, and a frequency of 28.90 kHz. The following gates were set: gate 1 was forward scatter FSC-Area against side scatter SSC-Area, gate 2 was FSC-Width against FSC-Height, gate 3 was SSC-Width against SSC-Height, and gate 4 was 530/40 (488 nm) against 710/50 (561 nm). Likely only the cell body and the inner segment survived the sample preparation and the cell sorting, while the axon and axon terminal are probably lost during the process. However, we argue that this does not affect the observed gene expression switch at P6. The reason is, that axons and axon terminals develop several days before the switch, at P1 and P3, respectively (this is visible in *Figure 1B*). In addition, in sensory cells, only a small fraction (6–10%) of the total amount of transcripts is localized to the axon (*Gumy et al., 2011*; *Taylor et al., 2009*; *Willis et al., 2005*). Taken together, the removal of axon-localized mRNA could have resulted in a small distortion of the observed expression pattern, but this distortion was likely constant from P3 on. RNA isolation was done using the Pico Pure kit. Total RNA amplification was done using the NuGEn Ovation RNA-Seq System V2 followed by library preparation with the Illumian TruSeq Nano DNA Sample prep protocol. Samples were sequenced in the Illumina Hiseq 2500 in 50 cycles, producing single-end reads.

## RNA sequencing analysis

Analysis was performed using the statistical software R using the QuasR package (*Gaidatzis et al., 2015b*). Sequences were mapped to a reference mouse genome (GRCm38/mm10) using splicedAlignment = TRUE, which internally uses SpliceMap (*Au et al., 2010*) counted per gene (using known gene annotation from UCSC) and normalized to the gene length and library size (RPKM = reads per kilobase transcript and million reads in library). Outliers were excluded from the analysis by removing the highest of the 39 values for each gene. The mean of the technical triplicates was calculated and a threshold was set to remove noise. To find the optimal threshold, different thresholds were applied and a curve was fitted to the resulting data set. The skewness of the curve was used as a measure of normal distribution. The threshold that resulted in the skewness closest to zero was chosen (threshold 1.645 RPKM). For the analyses, RPKM values were $\log_2$ transformed. To avoid minus infinity values after the log transformation, a small pseudocount was added to all values that equaled zero (in 91 instances). The smallest number that was not zero was chosen and a random integer between 1 and 10 divided by $10^6$ was added. For the principal components analysis (PCA) the data were centered (mean subtracted) and the genes were considered as variables. Fold change was calculated as the mean $\log_2$ expression after the switch (P7 to P12) minus the mean $\log_2$ expression before the switch (P0 to P5). Switch-on genes were defined as having a $\log_2$ fold change of 1 or more, switch-off genes were defined as having a $\log_2$ fold change of $-1$ or less. Differential gene expression was tested using the likelihood-ratio test using EdgeR. Genes with an FDR of less than

5% were considered differentially expressed. Constant genes were defined as having $\log_2$ fold changes between 0.01 and −0.01. The information on the chromosomal location of each gene was retrieved from Ensembl Biomart (http://www.ensembl.org/biomart/) for the mouse genome GRCm38.p4. The expression level is the mean expression after the switch (P7 to P12) or mean expression before the switch (P0 to P5). For each gene, the higher of the two values was chosen. Cone-specific genes were defined chosen from a cell type transcriptome study (*Siegert et al., 2012*). The cellular pathways were obtained from the KEGG database (http://www.genome.jp/kegg/pathway.html). To analyze the frequency of transcription factors among up- and downregulated genes, all expressed genes were ordered according to their fold changes, both ascending (for down-regulated genes) and descending (for upregulated genes), and binned to bins of 100, 200, 300, 400, 500, and 600 genes. The observed frequency in each bin was compared to the expected frequency by a permutation analysis. Sets of 100 genes were drawn randomly (1 million repetitions, with replacement) from all expressed genes, and the mean frequency of transcription factors among them was calculated. If the observed frequency occurred in less than 5% of the draws, a bin was considered to be significantly enriched for transcription factors. Analogously, cone-specific genes and pathways were tested for being significantly up- or downregulated. For the analysis, cone-specific genes were treated as a pathway. The mean fold change was calculated for each pathway and compared to the mean fold change of sets of randomly drawn fold changes. The number of elements in each set was the same as in the analyzed pathway. To determine the number of draws needed, p-values were compared at $10^2$, $10^3$, $10^4$, $10^5$, and $10^6$ draws (*Figure 3—figure supplement 1*). They were found to be stable after $10^5$ draws, and $10^6$ draws were chosen for the analysis. The number of drawn fold changes equaled the number of genes in a given pathway. A pathway was considered to be significantly up- or downregulated if 5% or fewer of the means of the randomly drawn fold changes were equal to or smaller than the absolute value of the observed fold change. If the observed fold change was not observed in any of the $10^6$ draws, the p-value was indicated as $p < 10^{-6}$. The exon-intron split analysis was performed as described in Gaidatzis *et al.* (*Gaidatzis et al., 2015a*). RNA sequencing reads obtained from exons and introns were quantified separately, and the change of exonic and intronic reads before and after the switch was correlated. The change was the difference between mean expression after the switch (P7 to P12) and mean expression before the switch (P0 to P5) in $\log_2$ space.

## ATACseq transposition assay library preparation and next generation sequencing

Both retinas from mice age P3, P6, and P9 were used for the ATACseq experiment. For each time point, three mice were used. Only male mice were used to avoid any influence from X-chromosome inactivation. The same procedure as for RNA sequencing was used for cell dissociation and FACS. Single-cell suspensions were obtained after FACS sorting into PBS/10%FCS. ATACseq was performed according to Buenrostro *et al.* (*Buenrostro et al., 2013*), with minor modifications. Briefly, 50,000 cells were used per transposition reaction. Cells were collected and lysed in cold lysis buffer (10 mM Tris-HCl pH 8.0; 10 mM NaCl; 3 mM $MgCl_2$; 0.5% NP40). Transposition was done as reported in Buenrostro *et al.* (*Buenrostro et al., 2013*), steps 6–12, and PCR amplification was done according to steps 13–18. Briefly, during the amplification reaction, PCR primers were used as mentioned in the supplemental material from Buenrostro et al. (*Buenrostro et al., 2013*). Amplified libraries were cleaned up using AMPure XP beads in a DNA:beads ratio of 1:1.8. Further size selection of the library prior to sequencing was avoided. The Nextera DNA Library Preparation Kit was used for library preparation. Multiple experiments were sequenced per lane (up to 3 per lane). Libraries were sequenced paired-end 100 bp reads using the Illumina HiSeq 2500.

## ATACseq analysis

Preprocessing of reads and analysis was performed using the statistical software R using the QuasR package (*Gaidatzis et al., 2015b*). Briefly, adaptor sequences (CTGTCTCTTATACACA) were removed from the 5'-ends of both reads using preprocessReads, and truncated reads were further trimmed by one more base to allow for paired-end alignment to a reference mouse genome (GRCm38/mm10) with qAlign with default parameters, which internally uses bowtie (*Langmead et al., 2009*). Around 70% of the obtained reads could be mapped. Peaks were

identified on a pool of all samples using macs2 version 2.1.0.20140616 (*Zhang et al., 2008*) using parameters –nomodel –shift 0 –gsize 1.87e9 –qvalue 0.10. The number of reads in each sample was then quantified in these peak regions. Mean counts for each time point were determined and $\log_2$ transformed (adding one count to each value to avoid minus infinity values). Outliers were removed by removing the ten highest peaks. The nearest transcription start site was found using the transcript annotation from the package TxDb.Mmusculus.UCSC.mm10.knownGene. The glmQLFit model (*Lund et al., 2012*) from the edgeR package (*McCarthy et al., 2012*) was used to determine differential peaks. Cutoff was at FDR 0.01. To compare our data to data from the literature (*Mo et al., 2016*), the regions of identified peaks in our data were chosen.

## Regularized regression model

The motif discovery analysis was done using an elastic net regression model (*Friedman et al., 2010*), as implemented in the glmnet package. The model predicts the $\log_2$ fold changes of accessibility in ATAC-seq peaks as a linear combination of the number of predicted transcription factor binding sites in each peak (=region of accessible chromatin). The coefficients (beta) obtained when fitting the model are then interpreted as the importance of each transcription factor for the observed differences in accessibility. Transcription factor binding motifs for 519 vertebrate transcription factors were obtained from the JASPAR2016 Bioconductor package, and binding sites were predicted in ATACseq peaks using a $\log_2$-oods cutoff of 10.0, or the maximal score of the weight matrix if it was below that value. In order to control for differences in peak width and sequence composition, three additional predictors were added to the transcription factor binding sites: peak width, the number of G + C bases and the number of CpG dinucleotides. Due to the similarities in the binding motifs of different transcription factors and the resulting correlation between the numbers of predicted binding sites per peak, fitting a classical linear model could result in uninformative coefficients. The elastic net used here is a form of regularized linear regression that does not suffer from this situation: it adds two additional penalization terms to the optimization function (their relative weight is controlled by parameter alpha, and their total weight by parameter lambda), which results in the beta coefficients of correlated predictors to be shrunk towards each other. In addition, coefficients for less important predictors are shrunk towards zero, resulting in a sparse result in which only important predictors have non-zero coefficients. Optimal values for alpha and lambda were obtained by a grid search: for each possible value of alpha between 0 and 1 in steps of 0.1, the optimal value of lambda was obtained by 5-fold cross-validation as the value of lambda that resulted in the minimal mean squared error. Finally, the optimal combination of both parameters was selected that resulted in the overall minimal mean squared error (alpha = 0.4, lambda = $5.169843e^{-5}$).

## Acknowledgements

We thank Christel Genoud for help with the serial block-face EM and Moritz Kischmann for help with the EM analysis software. We thank Stephane Thiry for help with RNA extraction and library preparation for RNA sequencing. We thank Tim Roloff and Kirsten Jacobeit for help with sequencing. We thank Brigitte Gross, Sabrina Djaffer, and Monique Lrech for technical support. We thank Jacek Krol, Stuard Trenholm, Santiago Rompani and Sara Oakeley for commenting on the manuscript. JD was supported by the Swiss National Science Foundation program 'NCCR Frontiers in Genetics'. BR was supported by Gebert-Rüf Foundation, Swiss National Science Foundation (3100330B_163457, CRSII3_141801), European Research Council (669157, RETMUS), National Centres of Competence in Research Molecular Systems Engineering, Swiss National Science Foundation Sinergia, Swiss-Hungarian, and European Union 3 × 3D Imaging grants.

## Additional information

### Funding

| Funder | Grant reference number | Author |
| --- | --- | --- |
| Swiss National Science Foundation Program 'NCCR Frontiers in Genetics' | | Janine M Daum |

| Swiss National Science Foundation | 3100330B_163457 | Botond Roska |
|---|---|---|
| European Research Council | 669157 | Botond Roska |
| Swiss National Science Foundation Sinergia | CRSII3_141801 | Botond Roska |
| European Research Council | RETMUS | Botond Roska |
| Gebert Rüf Stiftung | GRS-039/12 | Botond Roska |
| Swiss-Hungarian Cooperation Programme | SH/7/2/8 | Botond Roska |
| European Union 3 × 3D Grant | 323945 | Botond Roska |
| National Centres of Competence in Research Molecular Systems Engineering | | Botond Roska |

The funders had no role in study design, data collection and interpretation, or the decision to submit the work for publication.

### Author contributions

Janine M Daum, Conceptualization, Resources, Data curation, Software, Formal analysis, Validation, Investigation, Visualization, Methodology, Writing—original draft, Project administration, Writing—review and editing; Özkan Keles, Sjoerd JB Holwerda, Hubertus Kohler, Investigation, Methodology; Filippo M Rijli, Methodology; Michael Stadler, Conceptualization, Data curation, Software, Formal analysis, Supervision; Botond Roska, Conceptualization, Data curation, Formal analysis, Supervision, Funding acquisition, Writing—original draft, Project administration, Writing—review and editing

### Author ORCIDs

Janine M Daum (iD) http://orcid.org/0000-0001-9995-7117
Filippo M Rijli (iD) http://orcid.org/0000-0003-0515-0182
Botond Roska (iD) http://orcid.org/0000-0002-9559-1450

### Ethics

Animal experimentation: Animal procedures were done in accordance with standard ethical guidelines (European Communities Guidelines on the Care and Use of Laboratory Animals, 86/609/EEC) and were approved by the Veterinary Department of the Canton of Basel-Stadt.

### Decision letter and Author response

Decision letter https://doi.org/10.7554/eLife.31437.019
Author response https://doi.org/10.7554/eLife.31437.020

## Additional files

### Supplementary files

• Supplementary file 1. (a) Gene expression switch of cone-specific genes. List of genes defined as cone-specific (based on a cell type transcriptome study [*Siegert et al., 2012*]), ordered by fold change. (b) Transcription factors that switch off. List of transcription factors that switch off ($\log_2$ fold change >1), ordered by fold change. Only genes expressed above threshold (Materials and methods) are shown. (c) Mean percent change of switch pathways. Mean change in percent of all significantly up- or downregulated pathways. Pathways are ordered by p-value (at 1 million iterations).
DOI: https://doi.org/10.7554/eLife.31437.013

• Transparent reporting form
DOI: https://doi.org/10.7554/eLife.31437.014

### Major datasets

The following datasets were generated:

| Author(s) | Year | Dataset title | Dataset URL | Database, license, and accessibility information |
|---|---|---|---|---|
| Daum JM, Keles Ö, Kohler H, Stadler MB, Roska B | 2017 | RNA sequencing of developing cone photoreceptors | https://www.ncbi.nlm.nih.gov/geo/query/acc.cgi?acc=GSE97534 | Publicly available at the NCBI Gene Expression Omnibus (accession no: GSE97534) |
| Daum JM, Keles Ö, Kohler H, Stadler MB, Roska B | 2017 | ATAC sequencing of developing cone photoreceptors | https://www.ncbi.nlm.nih.gov/geo/query/acc.cgi?acc=GSE97535 | Publicly available at the NCBI Gene Expression Omnibus (accession no: GSE97535) |

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
