## [Decision Letter]

Thank you for submitting your article "The formation of the light-sensing compartment of cone photoreceptors coincides with a transcriptional switch" for consideration by *eLife*. Your article has been reviewed by two peer reviewers, and the evaluation has been overseen by a Reviewing Editor and Eve Marder as the Senior Editor. The following individual involved in review of your submission has agreed to reveal his identity: David Schaeffer (Reviewer #2).

The reviewers have discussed the reviews with one another and the Reviewing Editor has drafted this decision to help you prepare a revised submission. Happily, the reviewers were both supportive of publication and mainly have suggestions for clarification.

Summary:

In their manuscript Daum et al. analysed the formation of cone outer segments in mouse development at the morphological and gene expression level. Outer segments represent the cellular compartment in which light is converted into a biological signal and are therefore essential for proper vision. Furthermore, in contrast to the nocturnal mouse, human vision mainly depends on cone function providing daylight and high-acuity vision. By combining morphological analysis via fluorescence and serial block-face electron microscopy with RNAseq and chromatin data, the authors demonstrate a transcriptional switch at the start of outer segment formation at postnatal day 6. The presented data generated by state-of-the-art technology, though mainly descriptive, is of high interest and provides in depth knowledge in the field of photoreceptor development and might help in the development of therapies aiming to regenerate outer segments of cones for the treatment of blinding diseases.

Essential revisions:

1) In Figure 1 the authors claim to show the ultrastructure of cone outer segments (see figure legend), however, in the main text it is said, that actually rod and cone outer segments cannot be distinguished in the ultra-structural images. Given that the vast majority of photoreceptors within the mouse retina are rods (i.e. 97%) it is most likely that rod outer segments rather than cone outer segments are shown. Therefore, the figure legend and the y-axis legend in Figure 1 should be changed accordingly. Furthermore, if possible provide representative EM images using cone opsin immuno-gold labeling, maybe by correlative light and electron microscopy (CLEM), to unequivocally show cone outer segments.

2) In Figure 1—figure supplement 2 the provided quantifications suggest that the majority (>80%) of GFP+ cells are also opsin+. Please clarify the obvious difference to the data in Figure 1, where opsin+ outer segments are just detected from P5 onwards.

3) Figure 2—figure supplement 1: please use more distinct colors for the graphs of "Constant" and "All" for easier differentiation.

4) Due to the wave of maturation from central to peripheral retinal regions, the authors use only central regions for their analysis, however, a definition of "central region" is not provided. Please add this information.

5) In the figure legend of Figure 1 it is stated that quantification of outer segments using EM images was performed on n=1 mouse per day – does this mean one or both retinas?

6) The same group recently reported the importance of miR-182 and miR-183 for outer segment formation (Busskamp et al., 2014). Though the expression of microRNAs are not analysed in this study, their potential influence in this process should be discussed in light of the findings of this new study.

7) There has been a progressively increasing amount of work showing that mRNA species show very distinct localization patterns within neurons, such that the mRNA profile of the cell processes/axons can be quite independent of that of the soma. To what extent are the authors confident that they have captured the full cone cell body by FACS prior to mRNA-seq? They should discuss whether the seq data are biased by potential/likely loss of cell processes during tissue processing and FACS, at least at later times when the cells become more morphologically mature and elaborated. This could in principle affect the finding that genes involved in axon guidance are downregulated during the P5-6 switch. This is not a critical issue, but the authors should discuss the extent to which cell structure was maintained or not during cell harvesting, and whether this could represent a source of variability in the data.

---

## [Author Response]

1) In Figure 1 the authors claim to show the ultrastructure of cone outer segments (see figure legend), however, in the main text it is said, that actually rod and cone outer segments cannot be distinguished in the ultra-structural images. Given that the vast majority of photoreceptors within the mouse retina are rods (i.e. 97%) it is most likely that rod outer segments rather than cone outer segments are shown. Therefore, the figure legend and the y-axis legend in Figure 1 should be changed accordingly. Furthermore, if possible provide representative EM images using cone opsin immuno-gold labeling, maybe by correlative light and electron microscopy (CLEM), to unequivocally show cone outer segments.

We thank the reviewer for this important remark. At these early postnatal stages, it is not possible to distinguish rods and cones by morphology in EM. Clear morphological distinction is only possible after P21-P28 when rod nuclei have a fully inverted nuclear architecture (Solovei et al., Cell, 2009). As the reviewer points out, 97% of the photoreceptors are rods. Therefore, in our EM reconstructions, the vast majority of cells are rods. We changed the Figure Legend of Figure 1 and the y-axis in Figure 1 in the revised Figures, accordingly. Additionally, we adapted the text in the revised Results section to draw the readers’ attention to this limitation. However, we have clear evidence that cone outer segment formation starts at P6 from the experiments using light microscopy looking at opsin staining.

We are not familiar with the technique of immuno-gold labeling of cones. However, there is published data showing that soon after the time of the switch that we found to occur at P6, cone outer segments can be detected using cone opsin immuno-EM in the mouse retina (Keady et al., Molecular Biology of the Cell, 2011).

2) In Figure 1—figure supplement 2 the provided quantifications suggest that the majority (>80%) of GFP+ cells are also opsin+. Please clarify the obvious difference to the data in Figure 1, where opsin+ outer segments are just detected from P5 onwards.

We thank the reviewers for this comment. We used a mixture of two opsin antibodies for those experiments: one against short-wavelength-sensitive opsin (S-opsin) and another one for middle/long-wavelength-sensitive opsin (M/L-opsin). M/L-opsin is one of the switch genes, and thus only starts being expressed at P6. However, S-opsin is already expressed from P0 onwards. Therefore, we were able to detect S-opsin expression at the time points P0 till P5. At these early time points, S-opsin expression is confined to the cell body and inner segment. The quantifications in Figure 1—figure supplement 2 are therefore reflecting S-opsin labeling that we detected in the inner segment, for the time points before P6. After P6, the staining reflects S- and M/L-opsin labeling of outer segments. To clarify this, we added a sentence to the revised figure legend of Figure 1—figure supplement 2: “Before P5, opsin signal was weak and confined to the cell bodies and inner segments. This opsin signal likely reflects S-opsin, because S-opsin is expressed from P0 onwards while M/L-opsin only starts being expressed at P6.”

3) Figure 2—figure supplement 1: please use more distinct colors for the graphs of "Constant" and "All" for easier differentiation.

We changed the color for “All” to bright blue.

4) Due to the wave of maturation from central to peripheral retinal regions, the authors use only central regions for their analysis, however, a definition of "central region" is not provided. Please add this information.

We thank the reviewer for this note. We used the retina’s central disk of 3 mm in diameter (the center being the optic nerve). We added the information to the revised Materials and methods section in paragraphs Light microscopy, Sample preparation electron microscopy, and RNA sequencing.

5) In the figure legend of Figure 1 it is stated that quantification of outer segments using EM images was performed on n=1 mouse per day – does this mean one or both retinas?

We thank the reviewer for this comment. We clarified this in the revised Manuscript. One retina was used for each time point. We added that information to the revised figure legend of Figure 1.

6) The same group recently reported the importance of miR-182 and miR-183 for outer segment formation (Busskamp et al., 2014). Though the expression of microRNAs are not analysed in this study, their potential influence in this process should be discussed in light of the findings of this new study.

Following the advice of the reviewer, in the revised Discussion section we discuss the potential influence of miR-183/182 on outer segment formation. We added the following text.

“Are miRNAs involved in the switch? The sensory-cell-specific miRNA-182/183/96 cluster was implicated in outer segment growth. First, this cluster was shown to be necessary for outer segment maintenance in adult cone photoreceptors (Busskamp et al., 2014). Second, it was shown that the expression of the cluster starts at P10 (Krol et al., 2015) and, third, that knocking-out of these miRNAs results in shortened outer segments (Fan et al., 2017; Xiang et al., 2017). Taken together, these studies suggest that this miRNA cluster is not involved in initiating the formation of outer segments at P6, but that it is needed later in development, for maintenance or perhaps also for elongation of outer segments.”

7) There has been a progressively increasing amount of work showing that mRNA species show very distinct localization patterns within neurons, such that the mRNA profile of the cell processes/axons can be quite independent of that of the soma. To what extent are the authors confident that they have captured the full cone cell body by FACS prior to mRNA-seq? They should discuss whether the seq data are biased by potential/likely loss of cell processes during tissue processing and FACS, at least at later times when the cells become more morphologically mature and elaborated. This could in principle affect the finding that genes involved in axon guidance are downregulated during the P5-6 switch. This is not a critical issue, but the authors should discuss the extent to which cell structure was maintained or not during cell harvesting, and whether this could represent a source of variability in the data.

We thank the reviewer for drawing our attention to this issue. Likely only the cell body and the inner segment survived the sample preparation and the cell sorting, while the axon and axon terminal are likely lost during the process. However, we argue that this does not affect the observed gene expression switch at P6. The reason is, that axons and axon terminals develop several days before the switch, at P1 and P3, respectively (this is visible in Figure 1). In addition, in sensory cells, only a small fraction (6-10%) of the total amount of transcripts is localized to the axon (Gumy et al., 2011, Willis et al., 2005, Taylor et al., 2009). Taken together, the removal of axon-localized mRNA could have resulted in a small distortion of the observed expression pattern, but this distortion was likely constant from P3 on. We added this discussion to the revised Materials and methods section.